DATA RELEASE

# Multicellular, IVT-derived, unmodified human transcriptome for nanopore-direct RNA analysis

Caroline A. McCormick[1,†], Stuart Akeson[1,†], Sepideh Tavakoli[1], Dylan Bloch[1], Isabel N. Klink[1], Miten Jain[1,2,*] and Sara H. Rouhanifard[1,*]

1 Department of Bioengineering, Northeastern University, Boston, MA, 02115, USA
2 Department of Physics, Northeastern University, Boston, MA, 02115, USA

## ABSTRACT

Nanopore direct RNA sequencing (DRS) enables measurements of RNA modifications. Modification-free transcripts are a practical and targeted control for DRS, providing a baseline measurement for canonical nucleotides within a matched and biologically-derived sequence context. However, these controls can be challenging to generate and carry nanopore-specific nuances that can impact analyses. We produced DRS datasets using modification-free transcripts from *in vitro* transcription of cDNA from six immortalized human cell lines. We characterized variation across cell lines and demonstrated how these may be interpreted. These data will serve as a versatile control and resource to the community for RNA modification analyses of human transcripts.

**Subjects** Genetics and Genomics, Transcriptomics, Bioinformatics

# DATA DESCRIPTION

## Context

Nanopore direct RNA sequencing (DRS) has emerged as a method for analyzing native RNA strands based on ionic current disruptions during translocation through a biological pore. Deviations in the ionic current disruptions may be attributed to RNA modifications [1], such as pseudouridine (Ψ) [2], $N^6$-methyladenosine (m$^6$A) [3], and inosine (I) [4]. Oxford Nanopore Technologies developed basecallers capable of identifying modifications in RNA and DNA but limited in scope. Out of the 170+ known RNA modifications [5], only m$^6$A can be identified *de novo* with a basecaller [6]. For other modifications, negative controls and additional bioinformatic tools are necessary. These negative controls include but are not limited to enzymatic knockouts of RNA modification machinery [7], synthetic RNA controls [8–10], expected distribution pore models for computational analysis [11, 12], and genomic and transcriptomic templates for *in vitro* transcription (IVT)-based negative controls (i.e., unmodified transcriptomes) [2, 13].

IVT-derived, unmodified transcriptomes [13, 14] are an attractive option for analyzing modifications using DRS [2, 13, 15–17]. Currently, m$^6$A is the only exception with several signal-focused tools, including Dorado [18], the Oxford Nanopore RNA base caller, and m6anet [6], a multiple instances learning-based neural network. To generate these IVT datasets, polyadenylated (poly-A) RNA strands are reverse transcribed to cDNA, PCR amplified, and then *in vitro* transcribed into RNA using canonical nucleotides. This process

**Submitted:** 03 April 2024

\* Corresponding authors. E-mail: mi.jain@northeastern.edu; s.rouhanifard@northeastern.edu

† Contributed equally.

Preprint submitted at https://doi.org/10.1101/2023.04.06.535889

maintains the sequence context of the initial poly(A) RNA sample while "erasing" the RNA modifications, providing a baseline to compare putative modification sites. However, IVT RNA derived from a single cell line may not comprehensively capture the landscape of expressed genes in the human transcriptome, for example, if applied to a different cell line. IVT RNA derived from multiple human cell lines could better capture these differences and be applied broadly.

We present a long-read, multicellular, poly-A RNA-based, IVT-derived, unmodified transcriptome dataset for DRS modification analysis. We identified and flagged positions where the IVT data set differs from the GRCh38 reference, which could result in false identifications of modification sites. We also propose a strategy for filtering out these sites. This includes several mismatch tolerance levels that an end user can select. We also created a pooled version of this IVT dataset for increased representation of genes and positions of interest at the cost of cell line specificity. Finally, we computed ionic current-level alignments for each cell line, allowing users to apply this dataset without additional preprocessing steps.

This publicly available dataset will be a resource to the direct RNA analysis community and help reduce the need for expensive IVT library preparation and sequencing for human samples. This strategy will serve as a framework for RNA modification analysis in other organisms.

## Methods
### Cell culture
HeLa (RRID:CVCL_0030), Hep-G2 (RRID:CVCL_0027), A-549 (RRID:CVCL_0023), and NTERA-2 (RRID:CVCL_0034) cells were cultured in Dulbecco's modified Eagle's medium (Gibco, 10566024) as a base; SH-SY5Y cells (RRID:CVCL_0019) were cultured in a base of 1:1 EMEM:F12; Jurkat cells were cultured in RPMI (SH30027FS, FisherScientific). All media was supplemented with 10% Fetal Bovine Serum (FB12999102, FisherScientific) and 1% Penicillin-Streptomycin (Lonza,17602E). Cells were cultured at 37 °C with 5% $CO_2$ in 10 cm tissue culture dishes until confluent.

### Total RNA extraction and Poly(A) selection
Total RNA extraction from cells and Poly(A) selection was performed using a protocol outlined previously [2]. Six 10 cm cell culture dishes with confluent cells were washed with ice-cold PBS and lysed with TRIzol (Invitrogen,15596026) at room temperature and transferred to an RNAse-free microcentrifuge tube. Chloroform was added to separate the total RNA in the aqueous supernatant from the organic phase containing DNA and cell debris below following centrifugation. The aqueous supernatant was then transferred to a fresh RNAse-free microcentrifuge tube, and an equal volume of 70% absolute ethanol was added. PureLink RNA Mini Kit (Invitrogen, 12183025) was used to purify the extracted total RNA in accordance with the Manufacturer's protocol. Total RNA concentration was measured using the Qubit™ RNA High Sensitivity (HS) assay (Thermo Fisher, Q32852).

Poly(A) selection was performed using NEBNext Poly(A) mRNA Magnetic Isolation Module (NEB, E7490L) according to the Manufacturer's protocol. The isolated Poly(A) selected RNA was eluted from the beads using Tris buffer. The poly(A) selected RNA concentration was measured using the same Qubit™ assay listed above.



**Table 1.** Primers used (see methods).

| Protocol Use | Oligo name | Sequence (5′ → 3′) |
|---|---|---|
| IVT Fwd Primer PCR | Nanopore_IVT_T7_Forward_Primer | TAATACGACTCACTATAGCGAGGCGGTTTTCTGTTGGTGCTGATATTGCT |
| IVT Rev Primer PCR | Nanopore_IVT_T7_Reverse_Primer | ACTTGCCTGTCGCTCTATCTTC |
| PCR F primer for Sanger seq at Chr1:23792793 | Sanger_Chr1:23792793_F | CCGTGTGGTGTATGTGTGGT |
| PCR R primer for Sanger seq at Chr1:23792793 | Sanger_Chr1:23792793_R | CAGGTAGCAGCCAAACAGGT |
| PCR F primer for Sanger seq at Chr2:117817639 | Sanger_Chr2:117817639_F | GGAGGCATGTCTCAAGAAGCA |
| PCR R primer for Sanger seq at Chr2:117817639 | Sanger_Chr2:117817639_R | AAACTAAATGGCTGAAGTTCAAAGA |
| PCR F primer for Sanger seq at Chr19:2917188 | Sanger_Chr19:2917188_F | ACTGTGGACGAAAAGCACCT |
| PCR R primer for Sanger seq at Chr19:2917188 | Sanger_Chr19:2917188_R | TCCGACACTGCTCGCATTT |
| PCR F primer for Sanger seq at Chr3:19950940 | Sanger_Chr3:19950940_F | GGACATGGCTAGTCGAGGC |
| PCR R primer for Sanger seq at Chr3:19950940 | Sanger_Chr3:19950940_R | AGAAAATCTCACCCCCAATGGT |
| PCR F primer for Sanger seq at Chr4:109816233 | Sanger_Chr4:109816233_F | ATGTCTTTTCGAGGCGGAGG |
| PCR R primer for Sanger seq at Chr4:109816233 | Sanger_Chr4:109816233_R | GGTCCTTGGTCTTGGCCTTT |
| PCR F primer for Sanger seq at Chr1:35603333 | PSMB2_PseudoU_F_Primer | TGTTTGGGTACCCTCTACCAC |
| PCR F primer for Sanger seq at Chr1:35603333 | PSMB2_PseudoU_R_Primer | AGGACATGATGTTAGGAGCCC |

### *In vitro transcription and polyadenylation*

The protocol for IVT, capping, and polyadenylation was described previously [2]. Briefly, the cDNA-PCR Sequencing Kit (SQK-PCS109) facilitated reverse transcription (RT) and strand switching (SS). VN and Strand-Switching primers were added to 100 ng of poly(A) selected RNA from the abovementioned step. cDNA was produced by Maxima H Minus Reverse Transcriptase (Thermo Scientific, EP0751). Using a thermocycler, the reaction protocol was as follows: RT and SS for 90 min at 42 °C (1 cycle), Heat inactivation for 5 min at 85 °C (1 cycle), and hold at 4 °C (∞). PCR amplification was performed using LongAmp Taq 2X Master Mix (NEB, M0287S) and the Nanopore_T7_IVT_Forward and Reverse primers. The thermocycling conditions were as follows: initial denaturation for 30 s at 95 °C (1 cycle), denaturation for 15 s at 95 °C (11 cycles), annealing for 15 s at 62 °C (11 cycles), extension for 15 min at 65 °C (11 cycles), a final extension for 15 min at 65 °C (1 cycle), and then an indefinite hold at 4 °C. The PCR products were treated with Exonuclease 1 (NEB, M0293S) to digest single-stranded products. The resulting product was purified using Sera-Mag Select beads (Cytiva, 29343045) according to the Manufacturer's protocol. IVT was performed on the purified PCR product using a HiScribe T7 High yield RNA Synthesis Kit (NEB, E2040S) and purified using a Monarch RNA Cleanup Kit (NEB, T2040S), both according to the Manufacturer's protocols. Polyadenylation was performed using *E. coli* Poly(A) Polymerase (NEB, M0276S) using 12 μg of input RNA, and EDTA was added to halt the reaction. The product was purified again using a Monarch RNA Cleanup Kit with a final elution volume of 12 μL. The concentration was taken using the Qubit™ RNA HS assay. All primers are listed in Table 1.

### *Sequencing, basecalling, and alignment procedure*

Each cell line was sequenced individually using Nanopore DRS on R9 flow cells with sequencing chemistry SQK002. DRS runs were base called with Guppy v6.4.2 (RRID:SCR_023196) using the high accuracy model and the default basecalling quality-score filter Q ≥ 7 [19]. Basecalled reads were aligned with minimap2 (v2.24 [20]; RRID:SCR_018550) to the GRCh38.p10 reference genome and Gencode.v45 transcript sequences:

Gencode.v45: minimap2 -ax map-ont

GRCh38.p10: minimap2 -ax map-ont -uf -k 14

Sequence Alignment Maps (SAMs) were filtered to include only primary alignments for downstream analysis:

samtools view -h -F 4 -F 256 -F 2048.

### NanoPlot [21]
Nanoplot was used to generate sequencing and gencode alignment stats:

NanoPlot --raw.

### NanoCount [22]
NanoCount (no options) was used to calculate transcript abundance. Basecalled reads were aligned to Gencode.v45 transcript sequences similarly as above with the additional option (-N 1) and not filtered further. HeLa biological DRS raw data was sourced from NIH NCBI-SRA BioProject: PRJNA777450.

### Genomic DNA extraction and sanger sequencing
We performed Sanger sequencing on HeLa genomic DNA (gDNA) to analyze putative mismatches. gDNA extraction was performed using a Monarch Genomic DNA Purification Kit (NEB, T3010S) following the Manufacturer's protocol for cultured cells with an input of 5e6 HeLa cells. PCR primers to amplify ~200 nucleotide regions surrounding the mismatches were designed using Primer-BLAST (RRID:SCR_003095) with default settings and checking primer specificity against the *Homo Sapiens* genome. These primers are listed in Table 1. Using the Manufacturer's protocol, the PCR reaction was set up with Q5 polymerase (NEB, M0491L). Thermocycling conditions were as follows: initial denaturation at 98 °C for 30 s, 25 cycles of 98 °C for 10 s, then 63 °C for 20 s and 72 °C for 15 s, final extension at 72 °C for 2 minutes, and holding at 10 °C. PCR products were purified using a Monarch PCR & DNA Cleanup Kit (NEB, T1030S) following the Manufacturer's protocol. The concentration of eluted DNA was determined using a Nanodrop. The purified PCR products were imaged on a 2% agarose Tris-borate-EDTA (TBE) gel to confirm specific amplification. Samples were sent to Quintara Biosciences for SimpliSeq™ Sanger sequencing.

### Mismatch analysis methods
To identify the scope of mismatches in the sequenced IVT data (Figure 1), we followed an align, pileup, and compare strategy. We used the alignments to the GRCh38.p10 reference genome.

A positional pileup for each IVT replicate was created with pysamstats [23].

pysamstats (--t variation)

The initial process of variant calling documented every location with a minimum of 10x coverage in the pooled cell line dataset with at least one called nucleotide differing from the reference. Subsequent filtering removed variants based on the number of occurrences of a given variant in relation to the number of canonical bases and deletions present at a location. We sequentially applied a filter of 30%, 40%, 50%, 60%, 70%, 80%, and 95% to the data.

Each set of variants was divided into three distinct bins: variants previously documented in Ensembl, variants that occurred in low-confidence 9mers, and novel variants for the IVT replicate set.

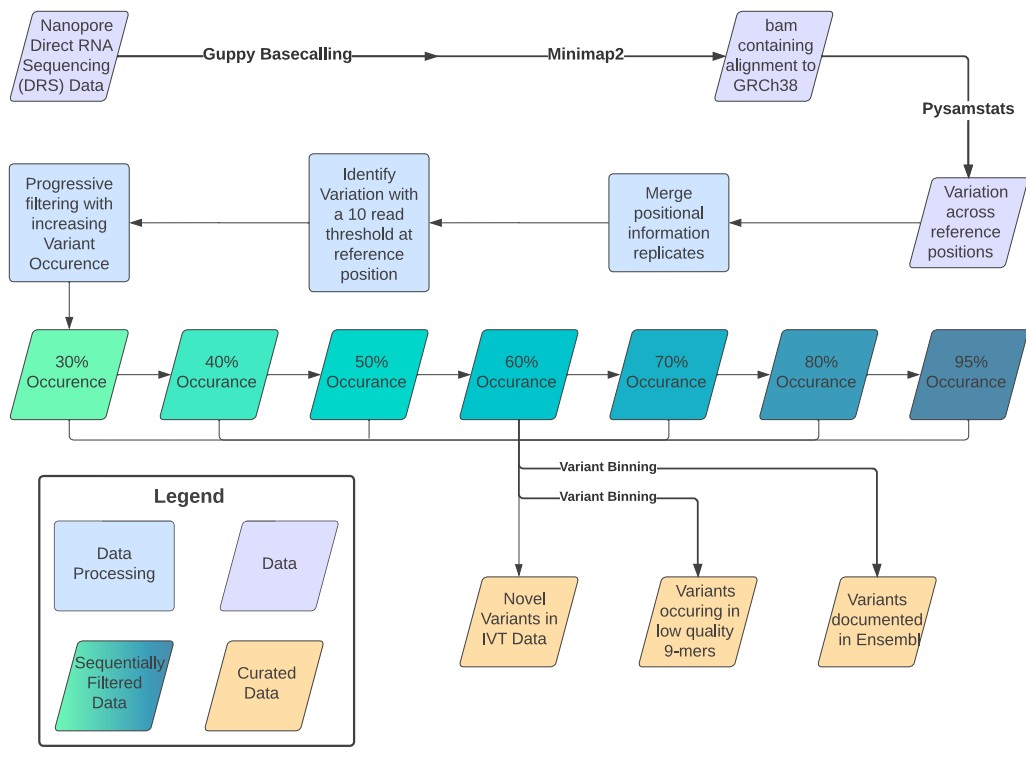

**Figure 1.** IVT to reference mismatch identification workflow.

The final dataset consisted of three compartments for each variant presence threshold: a set of potentially novel IVT variants, variants with known documentation in Ensembl, and variants that occurred in low-confidence 9mers [24].

### Round robin gene coverage saturation

To calculate the degree to which five of our cell lines (representative population) could approximate the observed gene set of the sixth one, we subsampled 1,000,000 reads from the sixth cell line (target cell line) to create a representation of the covered genes of that cell line. For each of the five other cell lines, we iteratively subsampled between 0 and 1,000,000 reads in increments of 100,000, for a combined total of 0 to 5,000,000 reads, and created a set of observed genes from those reads. At each iteration, we divided the cardinality of the union of our representative population with the target population by the cardinality of our target population to yield a proportion of the covered value. We repeated our sampling of the representative population 100 times and averaged the results for each sample interval. We repeated this process with each cell line acting as the target cell line. The iterative process was performed 500 times, yielding six saturation curves.

### Nanopolish eventalign

Eventalign [19] data for each dataset was computed with the following options

nanopolish eventalign --print-read-names --scale-events --samples

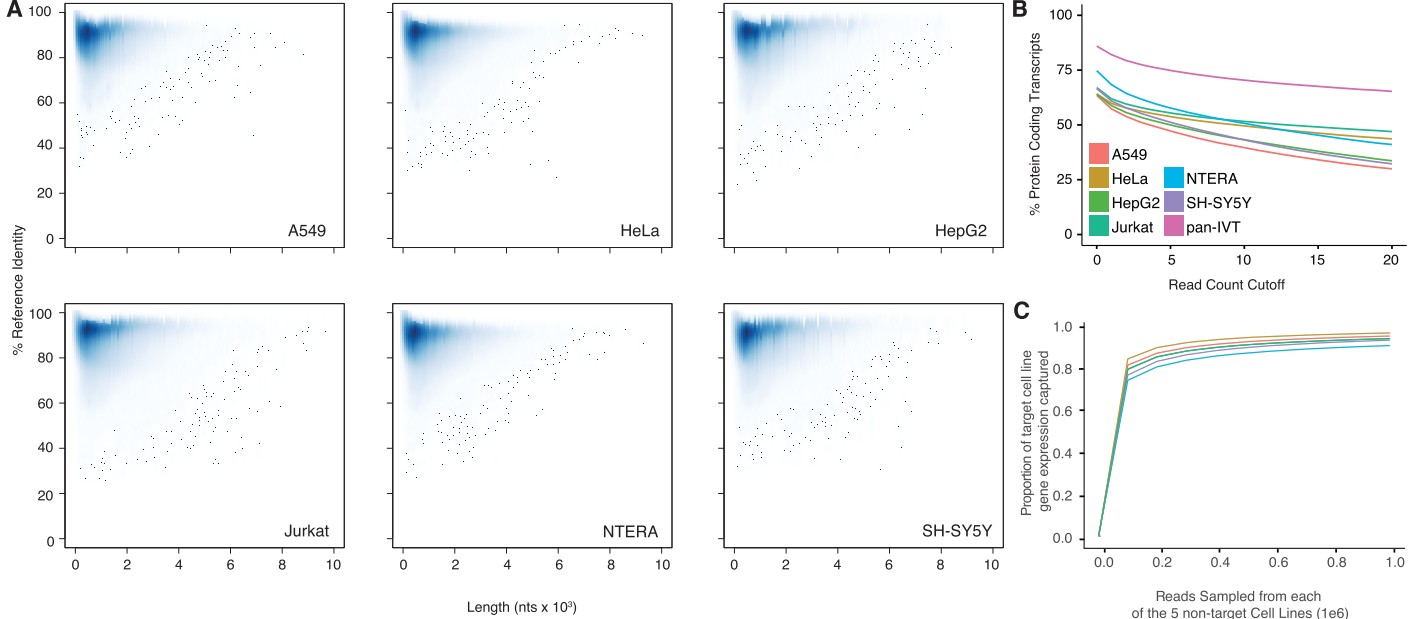

**Figure 2.** Alignment performance and coverage comparison. (A) Average percent identity of aligned DRS reads to Gencode.v38 transcript sequences against read length (nucleotides); data generated with Nanoplot [21]. (B) Percent of observed protein-coding transcripts (out of total protein-coding transcripts) against the minimum read-count cutoff of the transcripts for each cell line. panIVT is the additive combination of all cell line coverage. (C) Gene representation of the target cell line as a function of reads sampled from five population cell lines. Each cell line was used as the target cell line; the line in the graph corresponds to the representation of the target cell line listed in the legend.

**Table 2.** Cell type alignment statistics.

| Cell line | Human approximation | Aligned reads | Read N50 | Median alignment identity | Aligned genes (≥10 reads) |
|---|---|---|---|---|---|
| A549 | Lung Tissue | 1,464,177 | 922 | 89.9 | 8,133 |
| HeLa | Cervical | 3,868,698 | 983 | 90.6 | 10,002 |
| HepG2 | Liver Tissue | 1,790,636 | 1,197 | 90.7 | 8,812 |
| Jurkat | T Lymphocytes | 4,872,485 | 1,112 | 91.4 | 10,380 |
| NTERA | Testes | 2,351,140 | 935 | 90.1 | 10,308 |
| SH-SY5Y | Neurons | 1,340,875 | 1,084 | 90.1 | 8,873 |

## Data validation and quality control
### *Generation and characterization of panIVT*

We extracted RNA from six immortalized human cell lines: A549, HeLa, HepG2, Jurkat, NTERA, and SH-SY5Y cells and selected polyadenylated transcripts. These transcripts were reverse transcribed to cDNA, then *in vitro* transcribed back into RNA using canonical ribonucleotide triphosphates according to Tavakoli et al. (see Methods). Each library was prepared for sequencing using an ONT Direct RNA Sequencing kit and sequenced on a MinION or PromethION flow cell. The throughput for these data ranged from 1.3 million to 4.9 million primary aligned reads per cell line (Table 2). Basecalling was performed using Guppy 6.4.2 and alignment using minimap2 to Gencode v.45 (RRID:SCR_014966) human reference transcripts (Figure 2A, Table 2).

We pooled the aligned IVT reads from each cell line ("panIVT"). We observed 17,038 unique genes aligned with at least one DRS read from the panIVT comprising 85.69% of all human protein-coding genes in Gencode v45. As the read count cutoff increased, the



**Table 3.** Overlapping mismatches between all cell lines for the 30% occurrence threshold.

|  | Unique | In 1 other | In 2 others | In 3 others | In 4 others | In All |
|---|---|---|---|---|---|---|
| A549 mismatches | 18,733 | 5,121 | 3,986 | 3,119 | 3,091 | 3,813 |
| HeLa mismatches | 32,776 | 11,524 | 6,772 | 4,314 | 3,509 | 3,813 |
| HepG2 mismatches | 24,577 | 8,214 | 5,110 | 3,724 | 3,254 | 3,813 |
| Jurkat mismatches | 54,698 | 10,616 | 5,908 | 3,511 | 2,764 | 3,813 |
| Ntera mismatches | 30,492 | 9,996 | 6,536 | 4,264 | 3,571 | 3,813 |
| SH-SY5Y mismatches | 30,163 | 9,445 | 5,405 | 3,844 | 3,331 | 3,813 |

coverage of the human protein-coding genes decreased (Figure 2B). As the panIVT comprises all six cell lines, it maximizes coverage and provides a more comprehensive representation of human protein-coding genes. We tested the ability of the five cell lines to capture the gene level diversity of the sixth cell line for increasing subsample sizes (Figure 2C). NTERA has the lowest proportion of observed genes represented in the sample population at ~90%. Combining different cell lines captured most of the observed gene-level diversity for a cell line of interest.

### Cataloging sequence variations in IVT data using the GRCh38 reference

For an RNA modification analysis, it is essential to distinguish systematic and spurious mismatches in IVT data. Systematic mismatches could be caused by genomic variation, IVT errors (i.e., due to reverse transcriptase or polymerase), sequencing-related errors, basecalling errors, or alignment errors. The sites that show these types of systematic errors in the IVT dataset should be omitted from RNA modification analyses. For each cell line, we identified positions with a mismatch percentage of 30% or higher from the GRCh38 reference genome. We further separated mismatches into one of three categories: known variants according to Ensembl [25], mismatches occurring in low confidence 9-mers (i.e., 9-mer regions where the basecaller is systematically less confident), and a combination of the remaining mismatches (Figure 1; see table in GigaDB [24]). The low confidence 9-mer set was curated by averaging the phred score base quality of all 9-mers in an independent biological DRS dataset [13] and selecting the lowest quartile. We identified 62,708 mismatches in our HeLa dataset using a minimum read count of 10 and a mismatch occurrence threshold of 30%. Of these 24,879 known variants, 8,930 originated in low-confidence 9-mers, leaving the remaining 28,899 mismatches [24]. Without further orthogonal investigation, we recommend excluding all positions with a mismatch percentage at the preselected threshold, regardless of the mismatch category. A tabulated version of the mismatch analysis findings for each cell line in our IVT dataset, as well as a pooled version, is publicly available on GitHub in conjunction with the BED (Browser Extensible Data) files for each variance threshold for integrated genome viewing (IGV). The overlap of mismatches between each of the cell lines for the 30% cutoff threshold can be found in Table 3.

## Re-use potential
### Use of IVT dataset for downstream RNA modification analyses
We intend these data to be a negative control for biological DRS modification analyses. Transcript coverage and abundance correlation between a paired IVT mRNAs and its corresponding biological mRNA can assess the quality of the IVT as a negative control. For example, we compared the transcripts per million (TPM) [22] of IVT RNA from HeLa

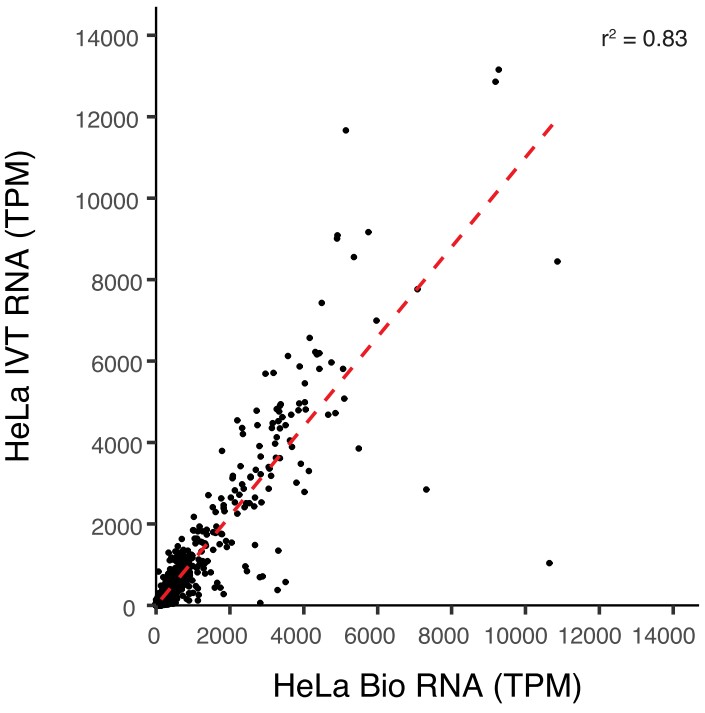

**Figure 3.** mRNA coverage (in TPM) correlation between HeLa IVT mRNA and HeLa biological mRNA.

cells to those of biological mRNA from HeLa cells, focusing on the aligned transcript [2]. The two TPM groups positively correlated ($r^2$ = 0.83), indicating that IVT is representative of the biological data (Figure 3). Figure 4A details an example decision process to determine whether a candidate DRS site should be included in the downstream analysis using this IVT dataset. The mismatched BED files can serve as a first filtration point for sites where a modification analysis is not recommended. These sites could include RNA editing sites, kmers that cause uncertainty in basecalling, or genetic variation that cannot be confirmed in the target sample as shown in Figure 4B. A stringent analysis would eliminate sites by using a 30% IVT mismatch filter. However, individual experiments may benefit from raising the occurrence threshold. We examined HeLa biological DRS data and BED files for 30%, 60%, and 95% occurrence thresholds in IGV (Figure 5A). Here, we can visualize positional nucleotide anomalies in HeLa biological DRS data. We can then refine the list of modification candidate sites by excluding positions where an IVT mismatch occurs at a given occurrence threshold (Figure 5B). If a particular site is being examined where a reference mismatch occurs (either IVT or biological), we recommend using orthogonal methods to confirm IVT as an appropriate negative control at that position. This should include but is not limited to sequencing gDNA from the same sample, as the direct RNA to confirm that the IVT identity and the DNA identity are the same. We recommend that biological conclusions drawn from analyses using these IVT data should include sufficient additional orthogonal confirmation.

We selected three example positions, determined their inclusion status (Figure 4A), and performed Sanger sequencing [26] as an orthogonal method to confirm our decision (Figure 4B). Two positions (chr2:117817639; chr1:23792793) showed a mismatch between the observed IVT sequence and the reference sequence (at 80% occurrence threshold);



**A**



**B**

Figure 4. Recommended Analysis Inclusion Criteria (A) Decision tree to determine if a position should be considered for downstream analysis. (B) Sanger sequencing for orthogonal support determines suitability for downstream modification analyses. Comparison of HeLa biological RNA (DRS), IVT RNA (DRS), gDNA (Sanger sequencing), and genome reference (GRCh38). Red bars indicate exclusion from downstream analyses, and green bars indicate inclusion.

hence, we recommend the third position (chr1:35603333) as a candidate site for downstream modification analyses.

At the first position of interest (chr2:117817639), both the biological RNA and IVT RNA were mismatched to the reference (GRCh38.p10). Sanger sequencing at chr2:117817639 revealed that the gDNA matched the biological and IVT RNA, indicating a single nucleotide variant (SNV). In this instance, we resolved to exclude this site as the biological RNA and IVT RNA agreed, strongly suggesting the absence of an RNA modification. Again, the biological and IVT RNA mismatched the reference nucleotide at the second position of interest (chr1:23792793). Unlike the preceding example, Sanger sequencing revealed that the gDNA matched the reference, indicating that the mismatch arose from a confounding variable. Therefore, this position was excluded from downstream analyses. The third position (chr1:35603333) represented a case where the IVT RNA matched the reference, but the biological RNA mismatched both the IVT RNA and the reference. Based on this information,

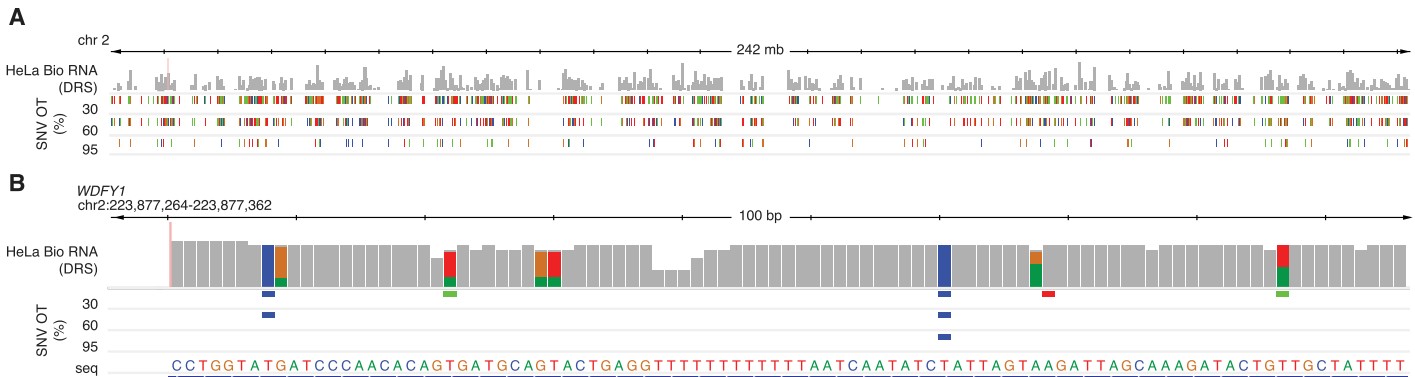

**Figure 5.** IGV visualization of HeLa DRS and BED files for 30%, 60%, and 95% SNV occurrence thresholds (OT). (A) View of chr2 displaying the read count depth of HeLa biological DRS (log scale) with corresponding IVT mismatch sites at each occurrence threshold. Colors indicate the mismatch presence of different nucleotides (red = T, green = A, blue = C, orange = G). (B) 100 nucleotide visualization of WDFY1 (chr2:223,877,264-223,877,362) where the reference nucleotide from hg38 genome is displayed as seq. Alignment mismatches for HeLa biological DRS are visualized proportionally as nucleotide count for respective colors. Grey indicates no significant mismatch at that position. Known variants at each occurrence threshold are denoted using the color of the variant nucleotide at that position.

we default to include this position for downstream analyses. Sanger sequencing confirmed our decision, as the gDNA agreed with the IVT RNA and reference, indicating a candidate site for further RNA modification analyses.

Once the candidate positions are identified, various bioinformatic tools exist that leverage IVT as a negative control. Some tools, such as Mod-*p* ID [2], compare the rate of mismatches observed in biological and IVT data. For this application, the publicly available bam files can be used. Other tools, such as nanocompore [11] and xPore [12], require nanopolish (RRID:SCR_016157) eventalign data [19] computed from the raw sequencing files. Since eventalign is a computationally intensive program, we precomputed the transcriptomic (gencode.v45) eventalign data and the genomic (GRCh38.p10) eventalign data and made them publicly available [27]. Figure 6A shows ionic current distributions for an example of five overlapping 5-mers. Cell lines with low coverage for the target 5-mer had noisier ionic current distributions.

## CONCLUSION

Appropriate negative controls are critical for accurately detecting and characterizing RNA modifications using nanopore DRS. IVT-derived negative controls provide an unmodified rendition of the transcriptome, allowing for comparative RNA modification analyses. A single cell line's IVT transcripts may not fully capture the diversity of the human transcriptome, but expansion to multiple cell lines provides a more comprehensive representation.

We created IVT DRS data for six immortalized cell lines. We cataloged sites where the IVT dataset did not match the human reference genome. With this information, filtering out sites with potentially confounding underlying sequences and drawing more robust conclusions during comparative analysis is possible. The underlying sequence variation can come from several factors, including genomic variations introduced during the *in vitro* transcription process. Either source of error makes the position a poor candidate for comparative modification analyses. Because of this, we recommend eliminating IVT mismatch sites, regardless of their origin, from comparative analyses.

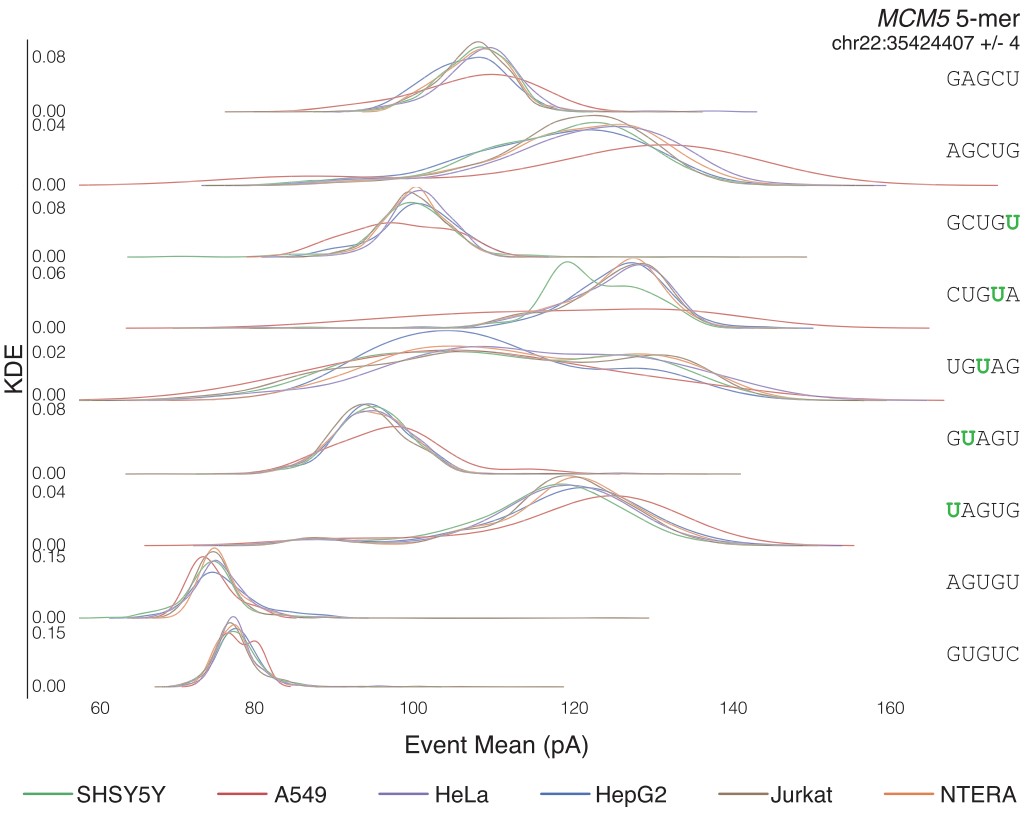

**Figure 6.** Ionic Current distributions for *MCM5* (chr22:35424407 +/− 4) across all six cell lines.

Once the candidate sites have been selected, bioinformatic tools can be used to perform comparative RNA modification analyses. We have precomputed and made publicly available nanopolish eventalign data to reduce the computational burden for potential users of this dataset. While these tools can have high entry barriers, we hope this dataset can help lower the computational burden and make RNA modification analyses more approachable for the community.

## AVAILABILITY OF SOURCE CODE AND REQUIREMENTS

- Project name: PanHumanIVT
- Project home page: https://github.com/RouhanifardLab/PanHumanIVT
- DOI: https://doi.org/10.5281/zenodo.7976171
- Operating systems: Linux
- Programming language: Python, R
- Other requirements: samtools 1.16.1 (using htslib 1.16), python 3.7, R 4.1.1, NanoPlot 1.40.2, Jupyterlab 3.4.4, NanoCount 1.0.0.post6
- License: MIT License.

## DATA AVAILABILITY

The datasets supporting the results of this article are available in the RouhanifardLab/PanHumanIVT GitHub repository [27]. Additional data and code snapshots are in Zenodo [28] and GigaDB [24].

FASTQ files and Fast5 raw data generated in this work have been made publicly available in NIH NCBI-SRA under the BioProject accession PRJNA947135.

Sequences were aligned to genome version hg38.p10 and gencode version 45 transcript sequences are available at GENCODE [29].

## ABBREVIATIONS

BED: Browser Extensible Data; DRS: direct RNA sequencing; gDNA: genomic DNA; HS: High Sensitivity; I: inosine;: IGV: integrated genome viewing; IVT: in vitro transcription; m6A: N6-methyladenosine; panIVT: pooled aligned IVT reads; RT: reverse transcription; SAM: Sequence Alignment Map SNV: single nucleotide variant; SS: strand switching; TPM: transcripts per million; Ψ: pseudouridine; TBE: Tris-borate-EDTA.

## DECLARATIONS

### Ethics approval and consent to participate

The authors declare that ethical approval was not required for this type of research.

### Competing interests

The authors declare no competing interests.

### Authors' contributions

CAM and SHR conceived the research. CAM and SA designed the experiments. CAM, ST, DB, and INK performed experiments and sequencing runs. CAM, SA, DB, and INK analyzed the data with guidance from SHR and MJ. CAM and SA wrote the paper with guidance from SHR and MJ.

### Funding

SHR acknowledges support from NIH 5R01HG011087, NIH R01HG012856, and support through an Opportunity Fund by the Technology Development Coordinating Center at Jackson Laboratories (NHGRI federal award no. U24HG011735).

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
