## [Reviewer Report]

Indicate in the comments box below whether you are happy with the changes made or if the manuscript is unacceptable.Comments on revised manuscriptI am happy to see the changes. Thanks!Indicate in the comments box below whether you are happy with the changes made or if the manuscript is unacceptable.Comments on revised manuscriptI am happy to see the changes. Thanks!

---

## [Editor Report]

Editor’s AssessmentOxford nanopore direct RNA sequencing (DRS) is a relatively new sequencing technology enabling measurements of RNA modifications. In vitro transcription (IVT)-based negative controls (i.e. modification-free transcripts) are a practical and targeted control for this direct sequencing, providing a baseline measurement for canonical nucleotides within a matched and biologically-derived sequence context. This work presents exactly this type of a long-read, multicellular, poly-A RNA-based, IVT-derived, unmodified transcriptome dataset. Review flagging more statistical analyses needed be performed for the data quality, and this was provided. The resulting data providing a resource to the direct RNA analysis community, helping reduce the need for expensive IVT library preparation and sequencing for human samples. And also serving as a framework for RNA modification analysis in other organisms.Editor’s AssessmentOxford nanopore direct RNA sequencing (DRS) is a relatively new sequencing technology enabling measurements of RNA modifications. In vitro transcription (IVT)-based negative controls (i.e. modification-free transcripts) are a practical and targeted control for this direct sequencing, providing a baseline measurement for canonical nucleotides within a matched and biologically-derived sequence context. This work presents exactly this type of a long-read, multicellular, poly-A RNA-based, IVT-derived, unmodified transcriptome dataset. Review flagging more statistical analyses needed be performed for the data quality, and this was provided. The resulting data providing a resource to the direct RNA analysis community, helping reduce the need for expensive IVT library preparation and sequencing for human samples. And also serving as a framework for RNA modification analysis in other organisms.

---

## [Reviewer Report]

Reviewer name and names of any other individual's who aided in reviewer Joshua T. BurdickDo you understand and agree to our policy of having open and named reviews, and having your review included with the published papers. (If no, please inform the editor that you cannot review this manuscript.)YesIs the language of sufficient quality?YesPlease add additional comments on language quality to clarify if needed
In line 284, "bioinformatic" may be more often used than "BioInformatic", but the meaning is clear.Are all data available and do they match the descriptions in the paper? YesAdditional CommentsAre the data and metadata consistent with relevant minimum information or reporting standards? See GigaDB checklists for examples <a href="http://gigadb.org/site/guide" target="_blank">http://gigadb.org/site/guide</a>YesAdditional CommentsPresumably the files (e.g. eventalign data) which are not in SRA will need to be uploaded to the GigaByte site.Is the data acquisition clear, complete and methodologically sound?YesAdditional CommentsIs there sufficient detail in the methods and data-processing steps to allow reproduction?YesAdditional CommentsLine 177 should presumably be "nanopolish evenetalign".Is there sufficient data validation and statistical analyses of data quality? YesAdditional CommentsIn my opinion, Figure 3(A) nicely illustrates the uncertainty in current nanopore data, which is useful.Is the validation suitable for this type of data?YesAdditional CommentsIs there sufficient information for others to reuse this dataset or integrate it with other data?YesAdditional CommentsAny Additional Overall Comments to the AuthorThe RNA samples, and nanopore sequencing data, should be useful as a negative control. Sequencing these IVT RNA samples using the newer ONT RNA004 pore and kit might also be useful.RecommendationAccept

---

## [Reviewer Report]

Reviewer name and names of any other individual's who aided in reviewer Jiaxu WangDo you understand and agree to our policy of having open and named reviews, and having your review included with the published papers. (If no, please inform the editor that you cannot review this manuscript.)YesIs the language of sufficient quality?YesPlease add additional comments on language quality to clarify if needed
Are all data available and do they match the descriptions in the paper? YesAdditional CommentsAre the data and metadata consistent with relevant minimum information or reporting standards? See GigaDB checklists for examples <a href="http://gigadb.org/site/guide" target="_blank">http://gigadb.org/site/guide</a>YesAdditional CommentsIs the data acquisition clear, complete and methodologically sound?YesAdditional CommentsIs there sufficient detail in the methods and data-processing steps to allow reproduction?YesAdditional CommentsIs there sufficient data validation and statistical analyses of data quality? NoAdditional CommentsThe authors ran DSR for the in vitro transcribed transcriptional RNAs from 6 cell lines to remove the possible natural modifications. The data can be used as a control RNA pool for natural or artificial modification studies.  however, more statistical analyses should be performed for the data quality. see comments below:   (1) For more possible usage of this data, some QC analysis is better to be provided to confirm the quality of these sequencing data. For example: 1) What is the correlation between in vitro transcribed transcriptional RNAs and original DSR for each cell line? 2) how many genes have been captured in each cell line?   (2) In Figure 2B, the author provides 3 conditions for ‘exclude’ and ‘include’, some statistical analysis should be performed to confirm how many cases in condition 1, condition 2, and condition 3. How many mismatches are showing in only 1 cell line, some cell lines or all the cell lines? The shared correct genes may be more confident references for the modification analysis.   (3) Different reads of the same gene could have different mismatches in the IVT RNAs due to RT-PCR bias or other reasons (especially for the lower expressed RNAs), for example, there are 100 reads in total, 90 reads are the correct nucleotide at a given position, 10 reads have a mismatch in the IVT sample, then how to define the signal as the control reference? Given that the nature modification is low in RNA, some threshold should be applied for the confident result, for example, what is the lowest expression threshold that could be used as a confident control reference? 
Is the validation suitable for this type of data?YesAdditional CommentsIs there sufficient information for others to reuse this dataset or integrate it with other data?NoAdditional CommentsFor more possible usage of this data, more QC data should be performed, please refer to my above commentsAny Additional Overall Comments to the AuthorRecommendationMajor Revision